# An Analytical Solution for Stress Transfer between a Broken Prestressing Wire and Mortar Coating in PCCP

**DOI:** 10.3390/ma15165779

**Published:** 2022-08-21

**Authors:** Xiaojie Zhang, Jiayu Wu, Chao Hou, Jian-Fei Chen

**Affiliations:** 1School of Astronautics, Harbin Institute of Technology, Harbin 150001, China; 2Department of Ocean Science and Engineering, Southern University of Science and Technology, Shenzhen 518000, China; 3Southern Marine Science and Engineering Guangdong Laboratory (Guangzhou), Guangzhou 511485, China

**Keywords:** prestress loss, stress transfer, fracture propagation, shear lag, PCCP, wire break

## Abstract

A prestressed concrete cylinder pipe (PCCP) consists of a concrete core, a steel cylinder, prestressing wires, and a mortar coating. Most PCCP failures are related to the breakage of prestressing wires. It is thus expected that the load-bearing capacity of PCCP is significantly affected by the length of the prestress loss zone and the stress distribution in the broken wire. Based on a tri-linear bond-slip model, the length of prestress loss zone and the stress transfer mechanism between a broken wire and a mortar coating are analysed in this paper. During the breaking (unloading) process of a prestressing wire, the interfacial bondline exhibits the following three stages: elastic stage, elastic-softening stage, and elastic-softening-debonding stage. The closed-form solutions for the interfacial slip, the interfacial shear stress, and the axial stress in the broken wire are derived for each stage. The solutions are verified by the finite element predictions. A parametric study is presented to investigate the effects of the size of the prestressing wires, the prestressing level, the interfacial shear strength, and the residual interfacial shear strength on the interfacial stress transfer. For an example PCCP with an inner diameter of 4 m, the length of prestress loss zone increases from 500 mm to 3300 mm as the radius of prestressing wire increases from 1 mm to 7 mm. It increases from 2700 mm to 7700 mm when the interfacial shear strength reduces from 3.94 MPa to 0.62 MPa and reduces from 13,200 mm to 7300 mm as the residual interfacial shear stress factor increases from 0.1 to 0.9.

## 1. Introduction

A prestressed concrete cylinder pipe (PCCP) consists of a concrete core, a thin steel cylinder, prestressing wires, and a mortar coating [1,2]. Due to its high strength and excellent durability, PCCP is widely used to transport and distribute water in municipal, industrial, and irrigation systems, such as in Libya’s Great Man-Made River project. There are two common types of PCCPs in practice: lined cylinder pipes (LCP) and embedded cylinder pipes (ECP). ECP has concrete encasement on both sides of the steel cylinder and circumferential prestressing wires wrap around the concrete core. A mortar coating is sprayed on the exterior of the prestressing wires protecting them from corrosion. However, the cracks may occur in the mortar coating over time leading to corrosion and breakage of the prestressing wires in harsh environments [3,4,5]. Hydrogen embrittlement may occur when ASTM A648 Class II steel wires, are embedded in PCCP, cathodically overprotected [6].

Once a prestressing wire breaks, the prestress in the concrete core reduces. This may lead to cracking in the concrete core under internal and/or external loading. Corrosive media may intrude through the cracks into the concrete core and corrode the steel cylinder and further damage the PCCP [7]. PCCP failures are usually catastrophic and can severely threaten public service and safety, such as in the case of a PCCP failure on River Road in Maryland and a PCCP water main break in Capitol Heights [8]. The location and number of broken wires can significantly affect the load-bearing capacity of PCCP, as evidenced by internal pressure and three-edge bearing tests [9], which may be detected using non-destructive testing methods such as acoustic emission [10,11], remote field eddy current/transformer coupling [12,13,14], acoustic fibre optic monitoring [15,16,17,18], and piezoceramics [19].

The finite element (FE) method has been widely used to numerically predict the load-bearing capacity of PCCP with broken wires. Some scholars [20,21] modelled the mortar coating, inner/outer concrete core, and steel cylinder of PCCP with a shell element, and modelled the prestressing effect using an equivalent radial pressure. This method is not suitable for PCCP with broken wires because it does not consider the stiffness of prestressing wires, the interfacial bond-slip behaviour, and the resulting local variation in the confining pressure. After prestressing wire breaks, the length of the prestress loss zone is a critical factor affecting the load-carrying capacity; limited research has been conducted to determine it. Ge [22], and Ge and Sinha [23] assumed that the bond strength was uniform in the prestress loss zone. Xiong [24] adopted an ideal elastoplastic model to simulate the interfacial behaviour, which does not consider the interface softening and debonding. Some scholars [25,26,27] did not consider the wrapping stress and the stiffness of the prestressing wire within the prestress loss zone, which is decided by conducting a wire break test.

Existing FE studies of PCCP with broken wires can be divided into two types in terms of the length of prestress loss zone: full prestress loss models [9,27,28,29,30,31,32,33,34] and partial prestress loss models [22,24,25,26,35]. In the former, the prestress is totally lost in a full wire wrap once a prestressing wire is broken and the wire wrap is removed. In the latter, the prestress is reduced locally within a certain arc length instead of one complete wire wrap. The length of prestress loss zone is usually important in establishing FE models of PCCP with broken wires. Experimental evidence [9,24,26,27] shows that the stress in a prestressing wire increases gradually from zero at the breaking point to the initial prestress beyond the prestress loss zone, due to the bond between the prestressing wire and the mortar coating. However, this nonuniform interfacial stress distribution cannot be obtained by using the existing FE models of PCCP with broken wires.

According to the best knowledge of the authors, few studies have focused on analytical solutions for the interfacial fracture between a prestressing wire and concrete (or mortar) in the literature. The mechanism of this interfacial stress transfer remains poorly understood. This paper presents a shear lag model applicable to the interfacial behaviour between a broken wire and mortar coating in an ECP (Figure 1). Based on the tri-linear bond-slip model, the governing equations are developed and solved. Analytical expressions for the slip, the interfacial shear stress, and the axial stress in the wire are obtained. FE simulations are conducted to verify the analytical solutions. Furthermore, a parametric study is presented to discuss the effects of the size of the prestressing wire, the prestressing level, the interfacial shear strength, and the residual interfacial shear strength factor on the interfacial shear stress transfer.

## 2. Assumptions and Governing Equations

Figure 1 shows a schematic of an ECP with a broken prestressing wire, which ruptures at breaking point A in Figure 1a. Due to the bond between the prestressing wire and mortar coating, the stress in the prestressing wire gradually recovers to the initial prestress at the prestress recovery Point B in Figure 1a. Arc AB in Figure 1a represents the length of the prestress loss zone on one side of the breaking point.

The equilibrium of a prestressing wire is illustrated in Figure 2. If the wrapping stress *f**_sg_* in the prestressing wire is constant, its relationship to the normal pressure *q_r_* acting on the concrete core is
(1)$$q_{r}=\frac{f_{\mathrm{sg}}A_{s}}{R_{s}}$$
where *R_s_* is the radius and *A_s_* is the cross-sectional area of the prestressing wire.

Figure 3 shows the forces on a prestressing wire after it breaks. The stress *σ**_s_* in the prestressing wire at breaking point A in Figure 3 drops to zero once the wire breaks. As the prestressing wire is embedded in the mortar coating, the bond between the wire and the mortar ensures that the stress *σ**_s_* in the prestressing wire recovers to the initial prestress at the prestress recovery point B in Figure 3. Within this prestress recovery zone, the interfacial normal pressure increases from zero at Point A to *q_r_* at Point B and there exists non-uniformly distributed interfacial shear stress as shown in Figure 3a. To obtain the stress in the wire, the interfacial shear, and normal stresses, a shear lag model is proposed in this study. The following assumptions are adopted in the theoretical derivation presented in this paper:The steel cylinder, mortar coating, and concrete core are regarded as rigid bodies because the stiffness of the pipe wall is much larger than the prestressing wires;The prestressing wires are thin and have negligible flexural stiffness;The pipe is large and there is sufficient bond between the wire and the mortar so that the breakage of a wire only results in local loss of the prestress;The wire remains linear elastic throughout the wire breaking process. All nonlinear deformation occurs at the wire-mortar interface.

In the polar coordinate system as shown in Figure 3, the strain in the broken wire at circumferential coordinate *θ* is given by
(2)$$\varepsilon_{s}=\frac{f_{\mathrm{sg}}}{E_{s}}-\frac{du_{s}}{\mathrm{ds}}$$
where *u**_s_* is the circumferential displacement of the broken wire.

AWWA C304-14 [1] recommends the use of the gross wrapping stress of the prestressing wire *f**_sg_* as 0.75*f**_su_*, and the yield strength of the prestressing wire *f**_sy_* as 0.85*f**_su_*; here, *f**_su_* is the specified tensile strength of the wires. The stress in the prestressing wire is
(3)$$\sigma_{s}=E_{s}\varepsilon_{s}=f_{\mathrm{sg}}-E_{s}\frac{du_{s}}{\mathrm{ds}}$$

Consider the arc AB as illustrated in Figure 3b. *F_loss_* (≤*f**_sg_A_s_*) is introduced to elucidate the prestress loss at point *A*. The process of wire breaking can be simulated as the prestress loss *F**_loss_* as the breaking point increases from 0 to *f**_sg_A_s_*. An infinitesimal body of the prestressing wire with an arc length of *ds* = *R_s_dθ* is shown in Figure 3c. The equilibrium equations of this infinitesimal body in the hoop and radial directions can be respectively expressed as follows:(4a)$$-2\pi r_{s}\tau \mathrm{ds}-T_{s}\cos\frac{d\theta }{2}+\left(T_{s}+dT_{s}\right)\cos\frac{d\theta }{2}=0$$
(4b)$$q_{b}\mathrm{ds}-\left(T_{s}+dT_{s}\right)\sin\frac{d\theta }{2}-T_{s}\sin\frac{d\theta }{2}=0$$
where *T**_s_* = *σ**_s_A_s_* is the tensile force in the prestressing wire. When *dθ* → 0, substituting $\cos\frac{d\theta }{2}\approx 1$ and $\sin\frac{d\theta }{2}\approx \frac{d\theta }{2}$ into Equation (4a,4b) yields
(5a)$$r_{s}\frac{d\sigma_{s}}{\;\mathrm{ds}}-2\tau =0$$
(5b)$$q_{b}=\frac{\sigma_{s}\pi r_{s}^{2}}{R_{s}}$$

The slip *δ* is defined as the relative displacement between the prestressing wire and the mortar coating, which is given by $\delta =-u_{s}$. Combining Equations (3) and (5a) results in the following governing equation
(6)$$\frac{E_{s}r_{s}}{2}\frac{d^{2}\delta }{ds^{2}}-\tau =0$$
where
(7)$$\sigma_{s}=f_{sg}+E_{s}\frac{d\delta }{ds}$$

## 3. Bond-Slip Model

The cohesive bond behaviour between the prestressing wire and mortar coating may be described using various bond-slip models such as those shown in Figure 4. The tri-linear bond-slip model (Figure 4a) [36,37,38,39] is adopted in this study. There are four key parameters to describe this model: the interfacial shear strength *τ**_f_*, the slip corresponding to the interfacial shear strength *δ*_1_, the slip corresponding to the residual interfacial shear strength *δ**_f_*, and the residual interfacial shear strength *τ**_r_* = *kτ**_f_*, where *k* is the residual interfacial shear strength factor. It can be mathematically expressed as:(8)$$\tau (\delta )=\{\begin{array}{ll} \frac{\tau_{f}}{\delta_{l}}\delta & \;\mathrm{for}\;0⩽\delta ⩽\delta_{1}\\ \frac{k\tau_{f}\left(\delta -\delta_{1}\right)+\tau_{f}\left(\delta_{f}-\delta \right)}{\delta_{f}-\delta_{1}} & \;\mathrm{for}\;\delta_{1}<\delta ⩽\delta_{f}\\ k\tau_{f} & \;\mathrm{for}\;\delta >\delta_{f} \end{array}$$

In the case of *k* = 0, Equation (8) is reduced to the bi-linear bond-slip model (Figure 4b), which has no residual interfacial shear strength. In the case of *k* = 1, Equation (8) represents an ideal elastoplastic model (Figure 4c).

## 4. Analytical Solutions

When the tri-linear bond-slip model is defined for the interface between the prestressing wire and mortar coating, the slip, interfacial shear stress, and the axial stress in the prestressing wire can be obtained by solving the governing equation (Equation (6)). Figure 5 illustrates the evolution of the interfacial shear stress distribution. The interfacial bond-slip behaviour after a wire break can be divided into three stages: the Elastic (E) stage, the Elastic-Softening (E-S) stage, and the Elastic-Softening-Debonding (E-S-D) stage, as shown in Figure 5.

Figure 6 shows a typical prestress loss–displacement curve at the breaking point. Here Δ represents the slip (displacement) at *s* = 0. *F_loss_*_,*sof*_ and *F_loss_*_,*deb*_ are respectively the prestress loss when the interface softening and debonding occurs at the breaking point (*A* in Figure 3). The interface may enter one of the three stages (Figure 5) after a wire break. The process to determine the interface status is shown in Figure 7:If the prestress loss at the breaking point satisfies *F_loss_*_,*sof*_ > *f**_sg_A_s_*, the entire length of the interface remains elastic (E stage, Figure 5a and segment OO_1_ in Figure 6).Interface softening initiates at the breaking point (Figure 5b) when the prestress loss satisfies *F_loss_*_,*sof*_ = *f**_sg_A_s_* (point O_1_ in Figure 6).The prestress loss zone includes the elastic state zone and softening state zone (Figure 5c and segment OO_2_ in Figure 6) as the wire breaking progresses, with *F_loss_*_,*deb*_ > *f**_sg_**A**_s_* > *F_loss_*_,*sof*_.Interface debonding initiates at the breaking point (Figure 5d) when the prestress loss is *F_loss_*_,*deb*_ = *f**_sg_**A**_s_* (point O_2_ in Figure 6).If the prestress loss at the breaking point satisfies *F_loss_*_,*deb*_ < *f**_sg_**A**_s_* (segment OO_3_ in Figure 6), the prestress loss zone includes an elastic state zone, a softening state zone, and a debonding state zone (Figure 5e).

### 4.1. Elastic Stage

The entire interface is at the E stage when the interfacial slip at the breaking point is less than *δ*_1_ (Figure 5a and segment OO_1_ in Figure 6). This stage ends when the shear stress reaches the bond shear stress *τ**_f_* at *s* = 0 (Figure 5b and point O_1_ in Figure 6).

Let $\lambda_{1}=\sqrt{\frac{2\tau_{f}}{E_{s}r_{s}\delta_{1}}}$, substituting the first term of Equation (8) into Equation (6) yields
(9)$$\frac{d^{2}\delta }{ds^{2}}-\frac{2\tau_{f}}{E_{s}r_{s}\delta_{1}}\delta =0$$
with boundary conditions $T_{s}(s=0)=\sigma_{s}A_{s}=f_{sg}A_{s}-F_{loss\;}$, $T_{s}(s=\infty )=f_{sg}A_{s}$, which are alternatively expressed as ${\frac{d\delta }{ds}|}_{s=0}=-\frac{F_{loss\;}}{f_{sg}A}$, ${\frac{d\delta }{ds}|}_{s=\infty }=0$.

The expressions of the slip, axial stress in the wire, interfacial shear stress, and interfacial normal stress can be obtained by solving Equation (9) to give:(10a)$$\delta =\frac{F_{loss\;}}{E_{s}\lambda_{1}A_{s}}e^{-\lambda_{1}s}$$
(10b)$$\sigma_{s}=f_{sg}-\frac{F_{loss\;}}{A_{s}}e^{-\lambda_{1}s}$$
(10c)$$\tau =\frac{\tau_{f}F_{loss\;}}{E_{s}\lambda_{1}\delta_{1}A_{s}}e^{-\lambda_{s}s}$$
(10d)$$q_{b}=\frac{\pi r_{s}^{2}}{R_{s}}\left(f_{sg}-\frac{F_{loss\;}}{A_{s}}e^{-\lambda_{1}s}\right)$$

The slip at the breaking point is obtained by substituting *s* = 0 into Equation (10a):(11)$$\Delta =\frac{F_{loss\;}}{E_{s}\lambda_{1}A_{s}}$$

The interface enters the E-S stage when Δ = *δ*_1_. The prestress loss at the initiation of interface softening is:(12)$$F_{loss,sof}=E_{s}\lambda_{1}\delta_{1}A_{s}$$

Provided *F**_loss_*_,*sof*_ < *f**_sg_A_s_*, the initiation of interface softening occurs but the prestressing wire has not yet broken. The interfacial enters into the E-S stage (Figure 7). In contrast, if *F**_loss_*_,*sof*_ ≥ *f**_sg_A_s_*, the prestressing wire breaks at the E stage. Substituting *F**_loss_* = *f**_sg_A_s_* into Equation (10a–10c) gives
(13a)$$\delta =\frac{f_{sg}}{E_{s}\lambda_{1}}e^{-\lambda_{1}s}$$
(13b)$$\sigma_{s}=f_{sg}-f_{sg}e^{-\lambda_{1}s}$$
(13c)$$\tau =\frac{\tau_{f}f_{sg}}{E_{s}\lambda_{1}\delta_{1}}e^{-\lambda_{s}s}$$
(13d)$$q_{b}=\frac{\pi r_{s}^{2}}{R_{s}}\left(f_{sg}-f_{sg}e^{-\lambda_{1}s}\right)$$
Substituting *s* = 0 into Equation (13a) results in
(14)$$\Delta =\frac{f_{sg}}{E_{s}\lambda_{1}}$$

The length of the prestress loss zone, denoted by *L*, may be defined as the length from the breaking point to the point where the stress in the prestressing wire is 0.95 of the initial prestress. According to Equation (13b), the length of prestress loss zone is
(15)$$L=\frac{3}{\lambda_{1}}$$

### 4.2. Elastic-Softening Stage

The wire-coating interface enters the E-S stage when the slip at the breaking point is in excess of *δ*_1_. The location of the peak interfacial shear stress *τ**_f_* is moved towards the prestress recovery point, and the interface near the breaking point enters the softening state, as shown in Figure 5c. The arc length from the breaking point to the location of the interfacial shear strength is the length of the softening zone, denoted by *s*_1_ (Figure 5c,d). At the end of the E-S stage (Figure 5d and point O_2_ in Figure 6), the interfacial shear stress *τ* at the breaking point reaches the residual interfacial shear strength *τ**_r_*. The prestress loss–displacement curve in the E-S stage is segment OO_2_ in Figure 6.

Substituting the first and second terms of Equation (8) into Equation (6), the following equations for the elastic zone and softening zone can be respectively obtained:(16a)$$\frac{d^{2}\delta }{ds^{2}}-\lambda_{1}^{2}\delta =0\;\mathrm{when}\;0⩽\delta ⩽\delta_{1}$$
(16b)$$\frac{d^{2}\delta }{ds^{2}}+\frac{2(1-k)\tau_{f}}{E_{s}r_{s}\left(\delta_{f}-\delta_{1}\right)}\delta =\frac{2\left(\tau_{f}\delta_{f}-k\tau_{f}\delta_{1}\right)}{E_{s}r_{s}\left(\delta_{f}-\delta_{1}\right)}\;\mathrm{when}\;\delta_{1}<\delta ⩽\delta_{f}$$
where $\lambda_{2}^{2}=\frac{2(1-k)\tau_{f}}{E_{s}r_{s}\left(\delta_{f}-\delta_{1}\right)}$ and $a_{1}=\frac{2\left(\tau_{f}\delta_{f}-k\tau_{f}\delta_{1}\right)}{E_{s}r_{s}\left(\delta_{f}-\delta_{1}\right)}$. The boundary and continuity conditions at the E-S stage are as follows
(17a)$$\sigma_{s}=f_{sg}-\frac{F_{loss\;}}{A_{s}},\;\mathrm{at}\;s=0$$

(17b)*σ**_s_* is continuous, at *s* = *s*_1_

(17c)*δ* = *δ*_1_, at *s* = *s*_1_

The interfacial slip, axial stress in the prestressing wire, interfacial shear stress, and interfacial normal stress for the elastic zone (0 ≤ *δ* ≤ *δ*_1_ or *s*_1_ ≤ *s*) can be obtained by solving Equation (16a):(18a)$$\delta =\delta_{1}e^{-\lambda_{1}\left(s-s_{1}\right)}$$
(18b)$$\sigma_{s}=f_{sg}-E_{s}\lambda_{1}\delta_{1}e^{-\lambda_{1}\left(s-s_{1}\right)}$$
(18c)$$\tau =\tau_{f}e^{-\lambda_{1}\left(s-s_{1}\right)}$$
(18d)$$q_{b}=\frac{\pi r_{s}^{2}}{R_{s}}\left[f_{sg}-E_{s}\lambda_{1}\delta_{1}e^{-\lambda_{1}\left(s-s_{1}\right)}\right]$$

The solutions for the softening zone (*δ*_1_ < *δ* ≤ *δ**_f_* or 0 ≤ *s* ≤ *s*_1_) are
(19a)$$\delta =b_{2}\frac{\cos\left(\lambda_{2}s\right)}{\cos\left(\lambda_{2}s_{1}\right)}+\frac{F_{loss\;}}{E_{s}\lambda_{2}A_{s}}\frac{\sin\left[\lambda_{2}\left(s_{1}-s\right)\right]}{\cos\left(\lambda_{2}s_{1}\right)}+\frac{a_{1}}{\lambda_{2}^{2}}$$
(19b)$$\sigma_{s}=f_{sg}+E_{s}\lambda_{2}\left[-b_{2}\frac{\sin\left(\lambda_{2}s\right)}{\cos\left(\lambda_{2}s_{1}\right)}-\frac{F_{loss\;}}{E_{s}\lambda_{2}A_{s}}\frac{\cos\left[\lambda_{2}\left(s_{1}-s\right)\right]}{\cos\left(\lambda_{2}s_{1}\right)}\right]$$
(19c)$$\tau =b_{1}b_{2}\frac{\cos\left(\lambda_{2}s\right)}{\cos\left(\lambda_{2}s_{1}\right)}+b_{1}\frac{F_{loss\;}}{E_{s}\lambda_{2}A_{s}}\frac{\sin\left[\lambda_{2}\left(s_{1}-s\right)\right]}{\cos\left(\lambda_{2}s_{1}\right)}+b_{1}\frac{a_{1}}{\lambda_{2}^{2}}+\frac{\tau_{f}\delta_{f}-k\tau_{f}\delta_{1}}{\delta_{f}-\delta_{1}}$$
(19d)$$q_{b}=\frac{\pi r_{s}^{2}}{R_{s}}\left\{f_{sg}+E_{s}\lambda_{2}\left[-b_{2}\frac{\sin\left(\lambda_{2}s\right)}{\cos\left(\lambda_{2}s_{1}\right)}-\frac{F_{loss\;}}{E_{s}\lambda_{2}A_{s}}\frac{\cos\left[\lambda_{2}\left(s_{1}-s\right)\right]}{\cos\left(\lambda_{2}s_{1}\right)}\right]\right\}$$
where $b_{1}=\frac{(k-1)\tau_{f}}{\delta_{f}-\delta_{1}}$, $b_{2}=\delta_{1}-\frac{a_{1}}{\lambda_{2}^{2}}$.

As *σ**_s_* is continuous at *s* = *s*_1_, substituting *s* = *s*_1_ into Equation (18b) and Equation (19b), the following equation can be obtained
(20)$$-\lambda_{1}\delta_{1}=\lambda_{2}\left[-b_{2}\tan\left(\lambda_{2}s_{1}\right)-\frac{F_{loss\;}}{E_{s}\lambda_{2}A_{s}}\frac{1}{\cos\left(\lambda_{2}s_{1}\right)}\right]$$

Substituting *s* = 0 into Equation (19a) gives
(21)$$\Delta =\frac{b_{2}}{\cos\left(\lambda_{2}s_{1}\right)}+\frac{F_{loss\;}}{E_{s}\lambda_{2}A_{s}}\tan\left(\lambda_{2}s_{1}\right)+\frac{a_{1}}{\lambda_{2}^{2}}$$
The interface debonding initiates at the wire breaking point when Δ = *δ_f_* at *s* = 0. Substituting Δ = *δ_f_* into Equation (21) leads to
(22)$$\delta_{f}=\frac{b_{2}}{\cos\left(\lambda_{2}s_{1}\right)}+\frac{F_{loss\;}}{E_{s}\lambda_{2}A_{s}}\tan\left(\lambda_{2}s_{1}\right)+\frac{a_{1}}{\lambda_{2}^{2}}$$

Combining Equations (20) and (22) yields:(23)$$\delta_{f}=b_{2}\cos\left(\lambda_{2}s_{1}\right)+\frac{\lambda_{1}\delta_{1}}{\lambda_{2}}\sin\left(\lambda_{2}s_{1}\right)+\frac{a_{1}}{\lambda_{2}^{2}}$$
Once the interface begins to enter the debonding zone, the length of softening zone *s*_1_ can be obtained from Equation (23) by iteration. Once the length of the softening zone *s*_1_ is obtained, the prestress loss *F**_loss_*_,*deb*_ at the initiation of interface debonding can be obtained by solving Equations (20):(24)$$F_{loss,\;deb\;}=E_{s}\lambda_{2}A_{s}\left[\frac{\lambda_{1}\delta_{1}}{\lambda_{2}}\cos\left(\lambda_{2}s_{1}\right)-b_{2}\sin\left(\lambda_{2}s_{1}\right)\right]$$

If *F**_loss_*_,*deb*_ < *f**_sg_A_s_*, the wire breaks without the interface entering the E-S stage (Figure 7). In contrast, if *F**_loss_*_,*deb*_ ≥ *f**_sg_A_s_*, the prestressing wire breaks with the interfaces entering the E-S stage. When the prestressing wire breaks, *F**_loss_* = *f**_sg_A_s_*. Substituting it into Equation (20) yields
(25)$$-\lambda_{1}\delta_{1}=\lambda_{2}\left[-b_{2}\tan\left(\lambda_{2}s_{1}\right)-\frac{f_{sg}}{E_{s}\lambda_{2}}\frac{1}{\cos\left(\lambda_{2}s_{1}\right)}\right]$$
The unknown parameter *s*_1_ can be computed from Equation (25) by iteration.

When the prestressing wire breaks, substituting *F**_loss_* = *f**_sg_A_s_* into Equations (18) and (19) gives
(26a)$$\delta =\{\begin{array}{ll} b_{2}\frac{\cos\left(\lambda_{2}s\right)}{\cos\left(\lambda_{2}s_{1}\right)}+\frac{f_{sg}}{E_{s}\lambda_{2}}\frac{\sin\left[\lambda_{2}\left(s_{1}-s\right)\right]}{\cos\left(\lambda_{2}s_{1}\right)}+\frac{a_{1}}{\lambda_{2}^{2}} & \;\mathrm{for}\;0⩽s⩽s_{1}\\ \delta_{1}e^{-\lambda_{1}\left(s-s_{1}\right)} & \;\mathrm{for}\;s_{1}⩽s \end{array}$$
(26b)$$\sigma_{s}=\{\begin{array}{ll} f_{sg}+E_{s}\lambda_{2}\left[-b_{2}\frac{\sin\left(\lambda_{2}s\right)}{\cos\left(\lambda_{2}s_{1}\right)}-\frac{f_{sg}}{E_{s}\lambda_{2}}\frac{\cos\left[\lambda_{2}\left(s_{1}-s\right)\right]}{\cos\left(\lambda_{2}s_{1}\right)}\right] & \;\mathrm{for}\;0⩽s<s_{1}\\ f_{sg}-E_{s}\lambda_{1}\delta_{1}e^{-\lambda_{1}\left(s-s_{1}\right)} & \;\mathrm{for}\;s_{1}⩽s \end{array}$$
(26c)$$\tau =\{\begin{array}{ll} b_{1}b_{2}\frac{\cos\left(\lambda_{2}s\right)}{\cos\left(\lambda_{2}s_{1}\right)}+b_{1}\frac{f_{sg}}{E_{s}\lambda_{2}}\frac{\sin\left[\lambda_{2}\left(s_{1}-s\right)\right]}{\cos\left(\lambda_{2}s_{1}\right)}+b_{1}\frac{a_{1}}{\lambda_{2}^{2}}+\frac{\tau_{f}\delta_{f}-k\tau_{f}\delta_{1}}{\delta_{f}-\delta_{1}} & \;\mathrm{for}\;0⩽s<s_{1}\\ \tau_{f}e^{-\lambda_{1}\left(s-s_{1}\right)} & \;\mathrm{for}\;s_{1}⩽s \end{array}$$
(26d)$$q_{b}=\{\begin{array}{ll} \frac{\pi r_{s}^{2}}{R_{s}}\left\{f_{sg}+E_{s}\lambda_{2}\left[-b_{2}\frac{\sin\left(\lambda_{2}s\right)}{\cos\left(\lambda_{2}s_{1}\right)}-\frac{f_{sg}}{E_{s}\lambda_{2}}\frac{\cos\left[\lambda_{2}\left(s_{1}-s\right)\right]}{\cos\left(\lambda_{2}s_{1}\right)}\right]\right\} & \;\mathrm{for}\;0⩽s<s_{1}\\ \frac{\pi r_{s}^{2}}{R_{s}}\left[f_{sg}-E_{s}\lambda_{1}\delta_{1}e^{-\lambda_{1}\left(s-s_{1}\right)}\right] & \;\mathrm{for}\;s_{1}⩽s \end{array}$$

Substituting $\sigma_{s}=0.95f_{sg}$ and *s* = *L* into the second term of Equation (26b), the length of prestressing loss zone in the E-S stage can be derived as
(27)$$L=s_{1}+\frac{3}{\lambda_{1}}-\frac{1}{\lambda_{1}}\ln\frac{f_{sg}}{E_{s}\lambda_{1}\delta_{1}}$$

### 4.3. Elastic-Softening-Debonding Stage

Debonding initiates at the breaking point when the interfacial shear stress *τ* there reduces to the residual interfacial shear strength *τ**_r_*. As debonding propagates, the peak interfacial shear stress continues to move towards the prestress recovery point. The interfacial shear stress within the debonded zone arises from friction. Wire breaking test results [26,35] have shown that the mortar coating can usually provide sufficient anchorage for the broken wire if the prestressing wire is long enough. The interface can be divided into three stress zones in this case: the elastic zone, the softening zone, and the debonded zone. A typical shear stress distribution is shown in Figure 5e. The arc lengths of the debonded and softening zones are respectively *s*_2_ and *s*_3_ − *s*_2_ (see Figure 5e). The prestress loss–displacement curve in the E-S-D stage is segment OO_3_ in Figure 6.

Substituting Equation (8) into Equation (6), the governing equations for the E-S-D stage can be obtained as:(28a)$$\frac{d^{2}\delta }{ds^{2}}-\lambda_{1}^{2}\delta =0\;\mathrm{when}\;0⩽\delta ⩽\delta_{1}$$
(28b)$$\frac{d^{2}\delta }{ds^{2}}+\lambda_{2}^{2}\delta =a_{1}\;\mathrm{when}\;\delta_{1}<\delta ⩽\delta_{f}$$
(28c)$$\frac{d^{2}\delta }{ds^{2}}-\lambda_{3}^{2}=0\;\mathrm{when}\;\delta \geq \delta_{f}$$
where $\lambda_{3}=\sqrt{\frac{2k\tau_{f}}{E_{s}r_{s}}}$. The boundary and continuity conditions are as follows:(29a)$$\sigma_{s}=f_{sg}-\frac{F_{loss\;}}{A_{s}}\;\mathrm{at}\;s=0$$
*δ* = *δ**_f_* at *s* = *s*_2_(29b)
*δ* = *δ*_1_ at *s* = *s*_3_(29c)
*σ**_s_* is continuous, at *s* = *s*_2_ and *s* = *s*_3_.(29d)
The solutions for the elastic zone with 0 ≤ *δ* ≤ *δ*_1_ (or *s*_3_ ≤ *s*) are
(30a)$$\delta =\delta_{1}e^{-\lambda_{1}\left(s-s_{3}\right)}$$
(30b)$$\sigma_{s}=f_{sg}-E_{s}\lambda_{1}\delta_{1}e^{-\lambda_{1}\left(s-s_{3}\right)}$$
(30c)$$\tau =\tau_{f}e^{-\lambda_{1}\left(s-s_{3}\right)}$$
(30d)$$q_{b}=\frac{\pi r_{s}^{2}}{R_{s}}\left[f_{sg}-E_{s}\lambda_{1}\delta_{1}e^{-\lambda_{1}\left(s-s_{3}\right)}\right]$$
The solutions for the softening zone with *δ*_1_ < *δ* ≤ *δ**_f_* (or *s*_2_ ≤ *s* < *s*_3_) are
(31a)$$\delta =b_{2}\frac{\sin\left[\lambda_{2}\left(s_{2}-s\right)\right]}{\sin\left(\lambda_{2}L_{1}\right)}+b_{3}\frac{\sin\left[\lambda_{2}\left(s-s_{3}\right)\right]}{\sin\left(\lambda_{2}L_{1}\right)}+\frac{a_{1}}{\lambda_{2}^{2}}$$
(31b)$$\sigma_{s}=f_{sg}+E_{s}\lambda_{2}\left[b_{3}\frac{\cos\left[\lambda_{2}\left(s-s_{3}\right)\right]}{\sin\left(\lambda_{2}L_{1}\right)}-b_{2}\frac{\cos\left[\lambda_{2}\left(s_{2}-s\right)\right]}{\sin\left(\lambda_{2}L_{1}\right)}\right]$$
(31c)$$\tau =\frac{\left(\delta_{f}-k\delta_{1}\right)\tau_{f}}{\delta_{f}-\delta_{1}}+b_{1}\left\{b_{2}\frac{\sin\left[\lambda_{2}\left(s_{2}-s\right)\right]}{\left(\lambda_{2}L_{1}\right)}+b_{3}\frac{\sin\left[\lambda_{2}\left(s-s_{3}\right)\right]}{\left(\lambda_{2}L_{1}\right)}+\frac{a_{1}}{\lambda_{2}^{2}}\right\}$$
(31d)$$q_{b}=\frac{\pi r_{s}^{2}}{R_{s}}\left\{f_{sg}+E_{s}\lambda_{2}\left[b_{3}\frac{\cos\left[\lambda_{2}\left(s-s_{3}\right)\right]}{\sin\left(\lambda_{2}L_{1}\right)}-b_{2}\frac{\cos\left[\lambda_{2}\left(s_{2}-s\right)\right]}{\sin\left(\lambda_{2}L_{1}\right)}\right]\right\}$$
where $L_{1}=s_{2}-s_{3}$, $b_{3}=\delta_{f}-\frac{a_{1}}{\lambda_{2}^{2}}$. The solutions for the debonded zone of the interface with 0 ≤ *s* < *s*_2_ (or *δ* > *δ*_f_) are
(32a)$$\delta =\frac{1}{2}\lambda_{3}^{2}s^{2}-\frac{F_{loss\;}}{E_{s}A_{s}}s+\delta_{f}+\frac{F_{loss\;}}{E_{s}A_{s}}s_{2}-\frac{1}{2}\lambda_{3}^{2}s_{2}^{2}$$
(32b)$$\sigma_{s}=f_{sg}+E_{s}\left(\lambda_{3}^{2}s-\frac{F_{loss\;}}{E_{s}A_{s}}\right)$$
(32c)$$\tau =k\tau_{f}$$
(32d)$$q_{b}=\frac{\pi r_{s}^{2}}{R_{s}}\left\{f_{sg}+E_{s}\left(\lambda_{3}^{2}s-\frac{F_{loss\;}}{E_{s}A_{s}}\right)\right\}$$

The stress in the prestressing wire *σ**_s_* is continuous at *s* = *s*_3_. It is possible to obtain *L*_1_ = *s*_2_ − *s*_3_ by iteratively solving the following equation:(33)$$-\lambda_{1}\delta_{1}=\lambda_{2}\left[b_{3}\frac{1}{\sin\left(\lambda_{2}L_{1}\right)}-b_{2}\cot\left(\lambda_{2}L_{1}\right)\right]$$
As *σ**_s_* is continuous at *s* = *s*_2_, combining Equations (31b) and (32b) gives:(34)$$s_{2}=\frac{\lambda_{2}}{\lambda_{3}^{2}}\left[b_{3}\cot\left(\lambda_{2}L_{1}\right)-b_{2}\frac{1}{\sin\left(\lambda_{2}L_{1}\right)}\right]+\frac{F_{loss\;}}{E_{s}A_{s}\lambda_{3}^{2}}$$
Substituting *s* = 0 into Equation (32a) yields
(35)$$\Delta =\delta_{f}+\frac{F_{loss\;}}{E_{s}A_{s}}s_{2}-\frac{1}{2}\lambda_{3}^{2}s_{2}^{2}$$
where *s*_2_ can be obtained by solving Equation (34).

When the prestressing wire breaks, *F**_loss_* = *f**_sg_A_s_*. Substituting *F**_loss_* = *f**_sg_A_s_* into Equation (34) gives
(36)$$s_{2}=\frac{\lambda_{2}}{\lambda_{3}^{2}}\left[b_{3}\cot\left(\lambda_{2}L_{1}\right)-b_{2}\frac{1}{\sin\left(\lambda_{2}L_{1}\right)}\right]+\frac{f_{sg}}{E_{s}\lambda_{3}^{2}}$$
The unknown variable *s*_2_ can be obtained from Equation (36). The interfacial slip, the axial stress in the prestressing wire, the interfacial shear and normal stresses after wire breaks are obtained as follows:(37a)$$\delta =\{\begin{array}{ll} \frac{1}{2}\lambda_{3}^{2}s^{2}-\frac{f_{sg}}{E_{s}}s+\delta_{f}+\frac{f_{sg}}{E_{s}}s_{2}-\frac{1}{2}\lambda_{3}^{2}s_{2}^{2} & \;\mathrm{for}\;0⩽s<s_{2}\\ b_{2}\frac{\sin\left[\lambda_{2}\left(s_{2}-s\right)\right]}{\sin\left(\lambda_{2}L_{1}\right)}+b_{3}\frac{\sin\left[\lambda_{2}\left(s-s_{3}\right)\right]}{\sin\left(\lambda_{2}L_{1}\right)}+\frac{a_{1}}{\lambda_{2}^{2}} & \;\mathrm{for}\;s_{2}⩽s<s_{3}\\ \delta_{1}e^{-\lambda_{1}\left(s-s_{3}\right)} & \;\mathrm{for}\;s_{3}⩽s \end{array}$$
(37b)$$\sigma_{s}=\{\begin{array}{ll} f_{sg}+E_{s}\left(\lambda_{3}^{2}s-\frac{f_{sg}}{E_{s}}\right) & \;\mathrm{for}\;0⩽s<s_{2}\\ f_{sg}+E_{s}\lambda_{2}\left[b_{3}\frac{\cos\left[\lambda_{2}\left(s-s_{3}\right)\right]}{\sin\left(\lambda_{2}L_{1}\right)}-b_{2}\frac{\cos\left[\lambda_{2}\left(s_{2}-s\right)\right]}{\sin\left(\lambda_{2}L_{1}\right)}\right] & \;\mathrm{for}\;s_{2}⩽s<s_{3}\\ f_{sg}-E_{s}\lambda_{1}\delta_{1}e^{-\lambda_{1}\left(s-s_{1}\right)} & \;\mathrm{for}\;s_{3}⩽s \end{array}$$
(37c)$$\tau =\{\begin{array}{ll} k\tau_{f} & \;\mathrm{for}\;0⩽s<s_{2}\\ \frac{\left(\delta_{f}-k\delta_{1}\right)\tau_{f}}{\delta_{f}-\delta_{1}}+b_{1}\left\{b_{2}\frac{\sin\left[\lambda_{2}\left(s_{2}-s\right)\right]}{\sin\left(\lambda_{2}L_{1}\right)}+b_{3}\frac{\sin\left[\lambda_{2}\left(s-s_{3}\right)\right]}{\sin\left(\lambda_{2}L_{1}\right)}+\frac{a_{1}}{\lambda_{2}^{2}}\right\} & \;\mathrm{for}\;s_{2}⩽s<s_{3}\\ \tau_{f}e^{-\lambda_{1}\left(s-s_{3}\right)} & \;\mathrm{for}\;s_{3}⩽s \end{array}$$
(37d)$$q_{b}=\{\begin{array}{ll} \frac{\pi r_{s}^{2}}{R_{s}}\left[f_{sg}+E_{s}\left(\lambda_{3}^{2}s-\frac{f_{sg}}{E_{s}}\right)\right] & \;\mathrm{for}\;0⩽s<s_{2}\\ \frac{\pi r_{s}^{2}}{R_{s}}\left\{f_{sg}+E_{s}\lambda_{2}\left[b_{3}\frac{\cos\left[\lambda_{2}\left(s-s_{3}\right)\right]}{\sin\left(\lambda_{2}L_{1}\right)}-b_{2}\frac{\cos\left[\lambda_{2}\left(s_{2}-s\right)\right]}{\sin\left(\lambda_{2}L_{1}\right)}\right]\right\} & \;\mathrm{for}\;s_{2}⩽s<s_{3}\\ \frac{\pi r_{s}^{2}}{R_{s}}\left[f_{sg}-E_{s}\lambda_{1}\delta_{1}e^{-\lambda_{1}\left(s-s_{1}\right)}\right] & \;\mathrm{for}\;s_{3}⩽s \end{array}$$

The length of prestress loss zone can be obtained by substituting *σ**_s_* = 0.95*f**_sg_* and *s* = *L* into the third term of Equation (37b):(38)$$L=s_{3}+\frac{3}{\lambda_{1}}-\frac{1}{\lambda_{1}}\ln\frac{f_{sg}}{E_{s}\lambda_{1}\delta_{1}}$$

## 5. Finite Element Modelling

To verify the above analytical solution, the prestressing wire–mortar interface was modelled using the finite element method (FEM) with the commercial software Abaqus [40]. The geometrical and material parameters of the ECP in [24], as listed in Table 1, were adopted. The material adopted in the finite element model is linear elastic.

The ECP was treated as a plane strain problem, with a half ring modelled (Figure 8). The concrete core, steel cylinder, and mortar coating were modelled by using the four-node 2D bilinear plane strain element CPE4R. The prestressing wire was modelled using the two-node 2D truss element T2D2. The prestressing wire–mortar interface was modelled using a cohesive contact approach using the tri-linear bond-slip model. According to Ren et al. [39], the four bond-slip model parameters can be calibrated from the experimental load–displacement curve, but this is not available from the literature. The bond-slip relationship for hot rolled smooth bars as in CEB-FIP [37] was adopted: $\tau_{f}=0.3\sqrt{f_{\mathrm{cm}}}=1.8\;\mathrm{MPa}$, *δ*_1_ = 0.1 mm. The residual interfacial shear strength *τ**_r_* = 0.5*τ**_f_* = 0.9 MPa and the corresponding slip *δ**_f_* = 1.0 mm were adopted for defining the remaining parameters of the tri-linear bond-slip model.

In the FE simulation, the wire breaking process was simulated through the following steps: (i) generating the FE mesh for the PCCP pipe; (ii) deactivating all the mortar coating elements; (iii) applying the prestress to the prestressing wire using the temperature drop method; (iv) reactivating mortar coating elements; (v) removing the restraint applied at one end of the prestressing wire to simulate the wire breaking (Figure 8b). In the FE model, the coefficient of linear thermal expansion was assumed to be zero for all other materials, but 1.2 × 10^−5^ °C^−1^ for the prestressing wire. A temperature drop of 389.53 °C was applied to generate a prestress of 902.39 MPa.

A mesh-sensitive analysis was conducted to obtain a reasonable mesh. Five FE models with 280, 560, 1120, 2240, and 4480 elements were established. Figure 9 shows the predicted displacement at the breaking point from these models. The predicted displacement decreases as the number of elements increases (Figure 9). The difference in the predicted displacement between the model with 1120 elements and the model with 4480 elements is very close, with a difference of 2.6%. On balancing accuracy and computational efficiency, the model with 1120 elements was adopted for the cases reported in this paper.

Figure 10 compares the analytical and FE predictions of the prestressing wire stress, interfacial shear stress, and the prestress loss–displacement relationship. It is seen that the analytical results are in very close agreement with the FE predictions. As the distance increases away from the breaking point, the axial stress in the prestressing wire increases and finally reaches the initial prestress at the prestress recovery point at about 2000 mm (the circumferential length of one loop of the prestressing wire is 14,765 mm), where the interfacial shear stress reduces to zero (Figure 10b). Therefore, the length of the prestress loss zone is not an entire wire wrap, which is consistent with the wire breaking tests [26,35]. When the prestress loss is small, the interfacial behaviour is elastic (in the E stage). It is followed by the E-S stage and E-S-D stage as the prestress loss increases. Figure 10b shows typical interfacial shear stress distribution at these three stages. The corresponding prestress loss–displacement curve is shown in Figure 10c.

## 6. Parametric Study

Based on analytical solutions (Equations (13), (26) and (37)), the interfacial shear and normal stresses, axial stress in the prestressing wire, and interfacial slip are affected by the radius of prestressing wire, the prestressing level, the interfacial shear strength, and the residual interfacial shear strength factor. The effects of the radius of prestressing wire, the prestressing level, the interfacial shear strength, and the residual interfacial shear strength factor on the interfacial shear stress are illustrated in Figure 11, Figure 12, Figure 13 and Figure 14. The dimensions and material properties listed in Table 1 were adopted in these examples.

### 6.1. Effect of the Size of the Prestressing Wire

Figure 11 shows how the radius of prestressing wire affects the interfacial shear and normal stress distributions. The interface is in the E-S stage when the prestressing wire is thin (*r**_s_* = 1.0 mm). It is at the E-S-D stage for thicker prestressing wires (*r**_s_* = 2.0, 3.5, 5.0, and 7.0 mm). The interfacial normal pressure increases from zero at the breaking point to *q_r_* at the prestress recovery point, which is in agreement with Ge’s [22] and Xiong’s [24] analyses. In addition, the length of prestress loss zone increases from 500 mm to 3300 mm as the radius of the prestressing wire increases from 1.0 mm to 7.0 mm.

### 6.2. Effect of the Prestressing Level

Five prestressing levels (0.45*f**_su_*, 0.575*f**_su_*, 0.75*f**_su_*, 0.85*f**_su_*, and 0.95*f**_su_*) were chosen to explore the effect of the prestressing level on the interfacial shear and normal stress distributions as shown in Figure 12. When the prestressing wire breaks, the entire length of the interface is at the E stage if for low prestresses (0.45*f**_su_* or 0.575*f**_su_*), whereas it enters the E-S stage for higher prestresses (0.75*f**_su_*, 0.85*f**_su_*, and 0.95*f**_su_*). The interfacial normal pressure increases from zero at the breaking point to *q_r_* at the prestress recovery point without significant differences for all cases, which differs from Ge’s [22] results. The reason is that Ge [22] assumed the interfacial shear stress is uniformly distributed and equal to the interfacial shear strength *τ**_f_*.

### 6.3. Effect of the Interfacial Shear Strength

According to CEB-FIB (2010) [37], the interfacial shear (bond) strength *τ**_f_* are $0.1\sqrt{f_{cm}}$ and $0.3\sqrt{f_{cm}}$ for cold drawn and hot rolled prestressing wire respectively. Also, the interfacial shear strength *τ_f_* is taken as $0.52\sqrt{f_{cm}}$ by Geng et al. [38], and $0.64\sqrt{f_{cm}}$ by Feldman et al. [41]. For the example case with *f**_cm_* = 72.5 MPa, the interfacial shear strength is calculated to be 0.62 MPa, 1.85 MPa, 3.2 MPa, and 3.94 MPa, respectively. Given these four values as examples, Figure 13 shows the corresponding interfacial shear and normal stress distributions. The interfacial shear strength clearly has a significant effect on the shear normal stress transfer. The interface is in the E stage when the bond strength is high (3.94 MPa and 3.2 MPa) because the mortar coating can provide sufficient anchoring force for the broken wire. It enters the state-S stage as the interfacial shear strength decreases to 1.85 MPa, and the E-S-D stage in the case of small interfacial shear strength (0.62 MPa). The length of prestress loss zone increases from 2700 mm to 7700 mm when the interfacial shear strength decreases from 3.94 MPa to 0.62 MPa. The length of prestress loss zone increases with the reduction of the interfacial shear strength which is consistent with the results of Ge [22] and Xiong [24].

### 6.4. Effect of the Residual Interfacial Shear Strength Factor

For the tri-linear bond-slip model, the range of the residual interfacial shear strength factor is 0 < *k* < 1; *k* = 0.1, 0.3, 0.5, 0.7, and 0.9 were adopted to investigate its effect on the interfacial shear and normal stress distributions as shown in Figure 14.

The interface is at the E-S-D stage in all cases (Figure 14a), but the length of the debonding zone increases as *k* decreases. The length of the prestress loss zone decreases from 13,200 mm to 7300 mm as the residual interfacial shear strength factor increases from 0.1 to 0.9. Figure 14b shows that the interfacial normal stress increases from zero at the breaking point at a slower rate for smaller *k* values.

## 7. Conclusions

Prestressed concrete cylinder pipes (PCCPs) are widely used for transporting and distributing water. Their failure is often associated with the rupture of prestressing wires due to corrosion. The length of the prestress loss zone can significantly affect the load-carrying capacity of PCCP. This paper has presented a shear lag model for predicting the stress transfer and debonding propagation after a prestressing wire breaks in a PCCP pipe. The following conclusions can be drawn.

(1)Adapting a tri-linear bond-slip model for the prestressing wire–mortar coating bond behaviour, closed-form expressions for the axial stress in the prestressing wire, the prestress loss–displacement relation, and the interfacial shear and normal stress distributions have been derived for elastic (E), elastic-softening (E-S), and elastic-softening-debonding (E-S-D) stages of the interface. The solutions have been verified by a 2D plane strain FE model of a PCCP with a broken wire. These analytical solutions can be used to determine the mechanical state of a prestressing wire after it breaks in a PCCP.(2)Based on a parametric study, it has been found that the size of prestressing wire, the prestressing level, the interfacial shear (bond) strength, and the residual interfacial shear strength factor have significant effects on the interfacial shear and normal stress distributions.(3)The length of prestress loss zone increases as the radius of prestressing wire and residual interfacial shear strength increase, and the interfacial shear strength reduces. For an example PCCP with an inner diameter of 4 m, the length of prestress loss zone increases from 500 mm to 3300 mm as the radius of prestressing wire increases from 1 mm to 7 mm and increases from 2700 mm to 7700 mm when the interfacial shear strength reduces from 3.94 MPa to 0.62 MPa, but it reduces from 13,200 mm to 7300 mm as the residual interfacial shear stress factor increases from 0.1 to 0.9.

The analytical solutions presented in this paper provide a basis for further rigorous study on the effect of wire breaking on the load-bearing capacity of PCCP, more advanced PCCP design guidance, and a theoretical basis for wire breaking tests.

## Figures and Tables

**Figure 1 materials-15-05779-f001:**
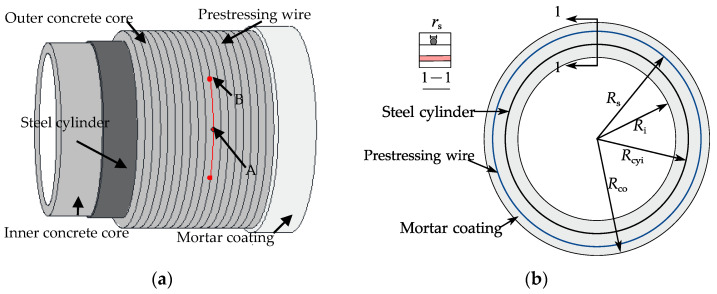
Schematics of (**a**) an ECP with broken wire and (**b**) a cross-section of ECP.

**Figure 2 materials-15-05779-f002:**
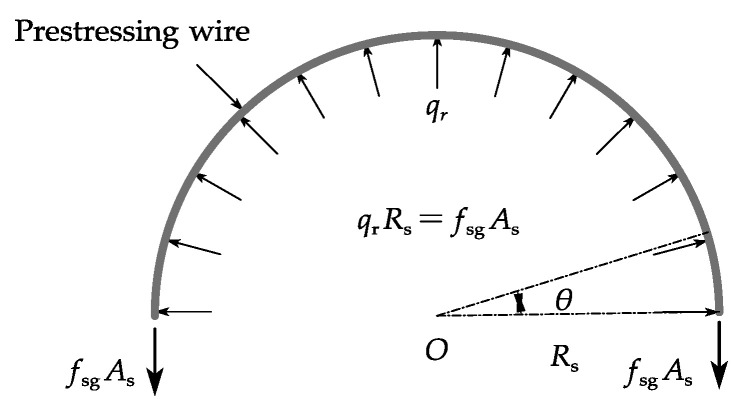
Equilibrium of a prestressing wire wrap.

**Figure 3 materials-15-05779-f003:**
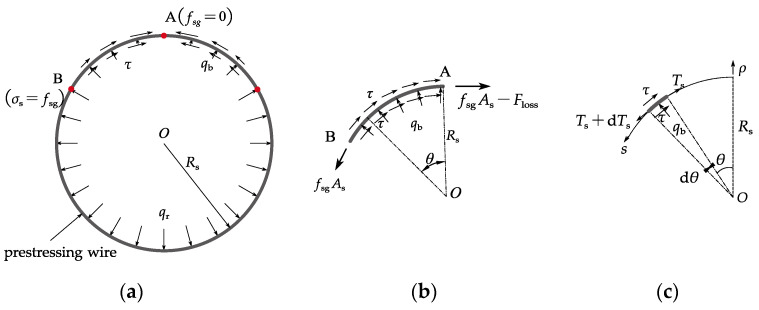
Force analysis diagram after a wire breaks: (**a**) Broken wire; (**b**) Stresses within the prestress loss zone; (**c**) An infinitesimal body.

**Figure 4 materials-15-05779-f004:**
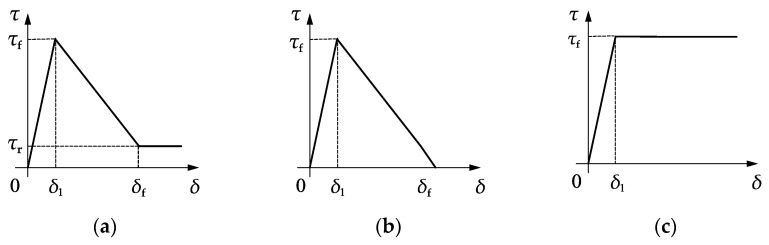
Bond-slip models: (**a**) Tri-linear model; (**b**) Bi-linear model; (**c**) Ideal elastoplastic model.

**Figure 5 materials-15-05779-f005:**
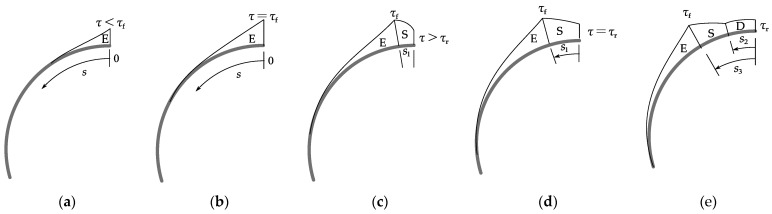
The evolution of interfacial shear stress distribution: (**a**) E stage; (**b**) End of E stage; (**c**) E-S stage; (**d**) End of E-S stage; (**e**) E-S-D stage.

**Figure 6 materials-15-05779-f006:**
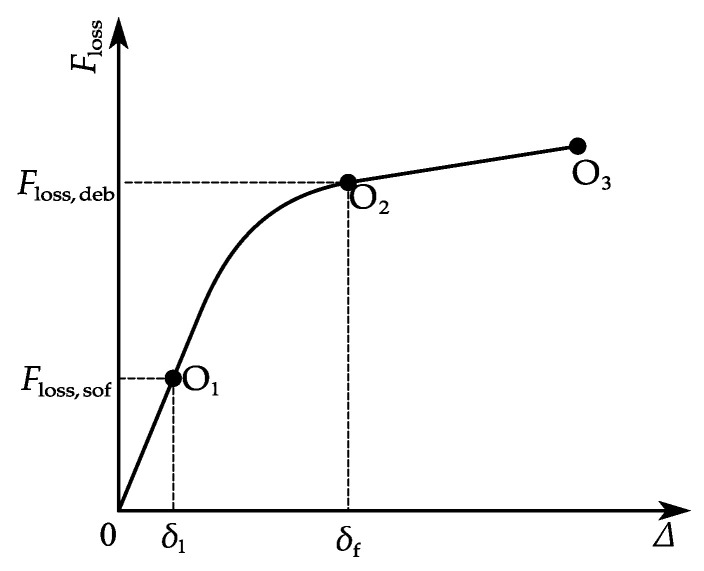
Typical prestress loss–displacement curve at the breaking point.

**Figure 7 materials-15-05779-f007:**
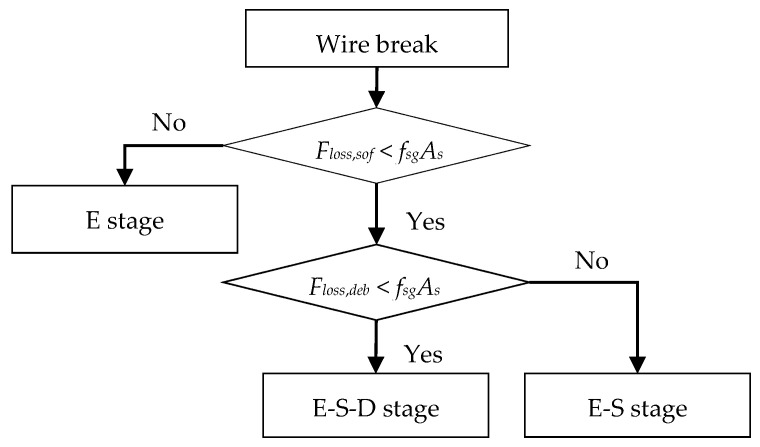
The process of determining the status of the interface after a wire break.

**Figure 8 materials-15-05779-f008:**
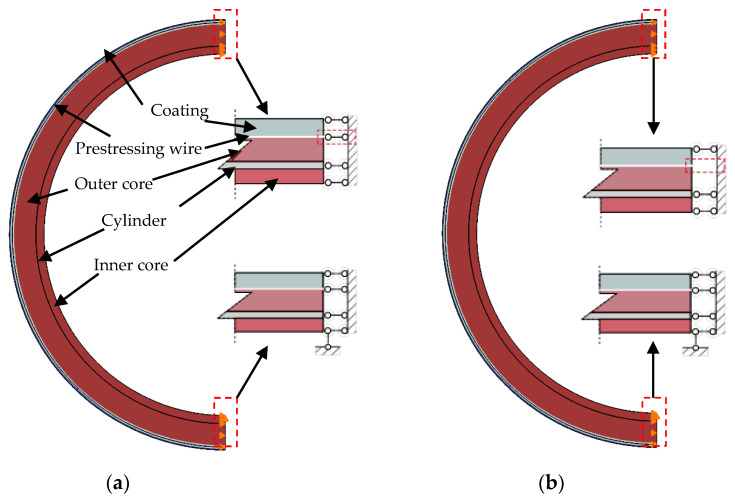
Boundary conditions for the FE model: (**a**) before wire breaks and (**b**) after wire breaks.

**Figure 9 materials-15-05779-f009:**
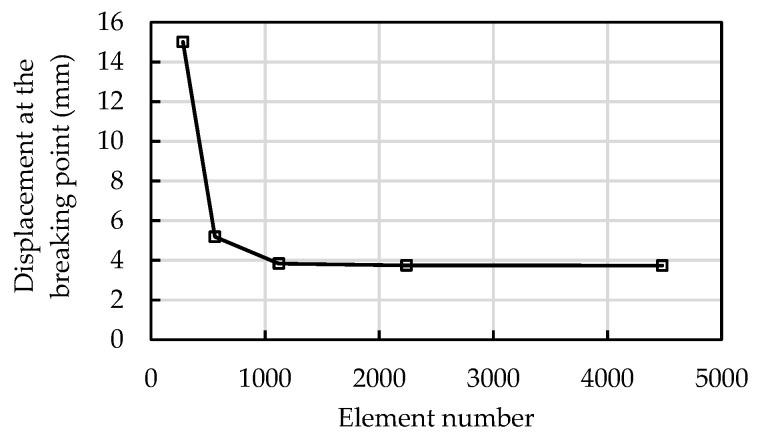
Mesh Convergence analysis.

**Figure 10 materials-15-05779-f010:**
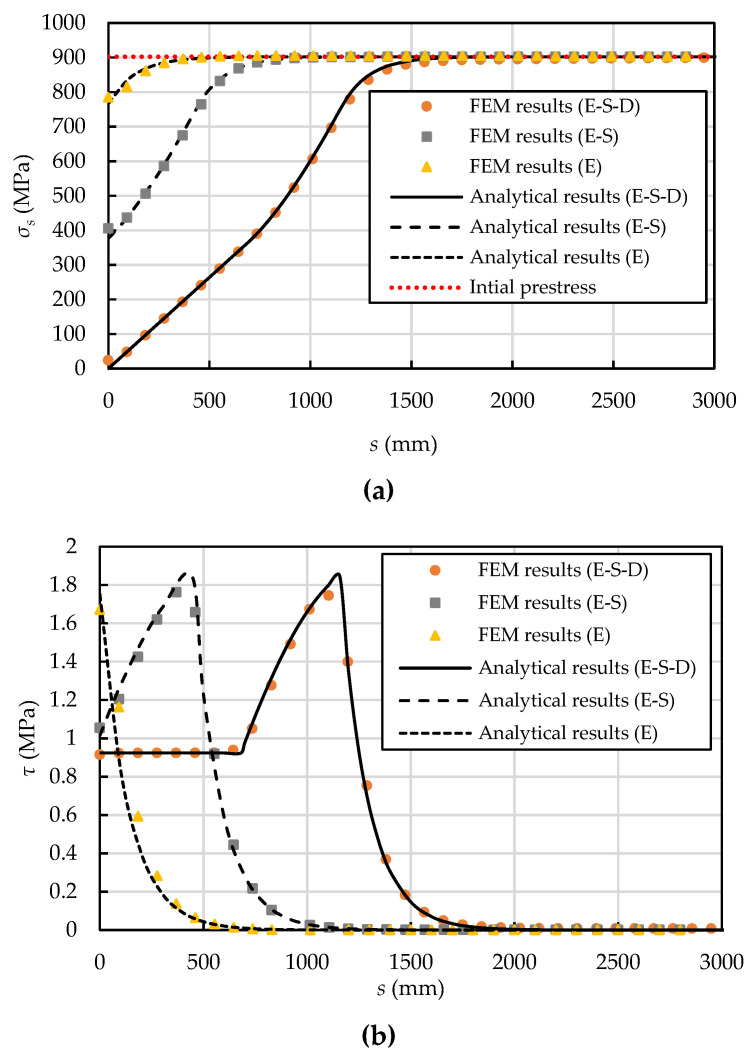
Comparisons of (**a**) the stress in the prestressing wire, (**b**) interfacial shear stress, and (**c**) prestress loss–displacement curve between analytical and FEM predictions.

**Figure 11 materials-15-05779-f011:**
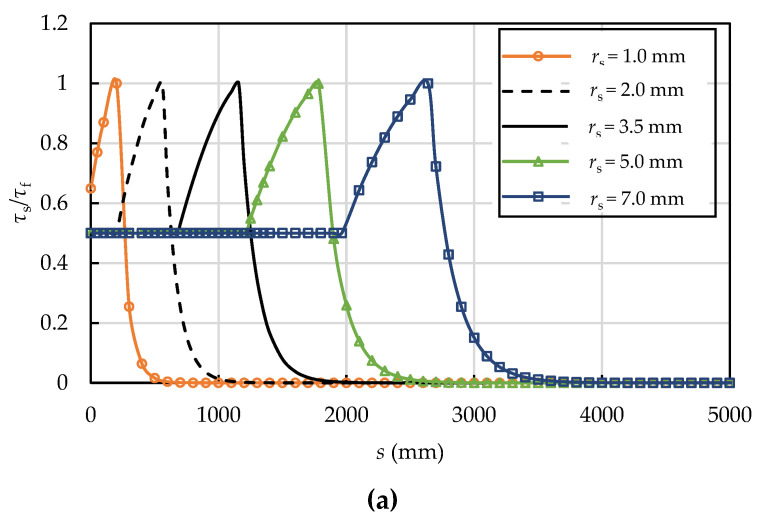
Effect of the size of the prestressing wire on the (**a**) interfacial shear and (**b**) normal stress distributions (*τ**_f_* = 1.8 MPa, *δ*_1_ = 0.10 mm, *k* = 0.5, and *δ**_f_* = 1.0 mm).

**Figure 12 materials-15-05779-f012:**
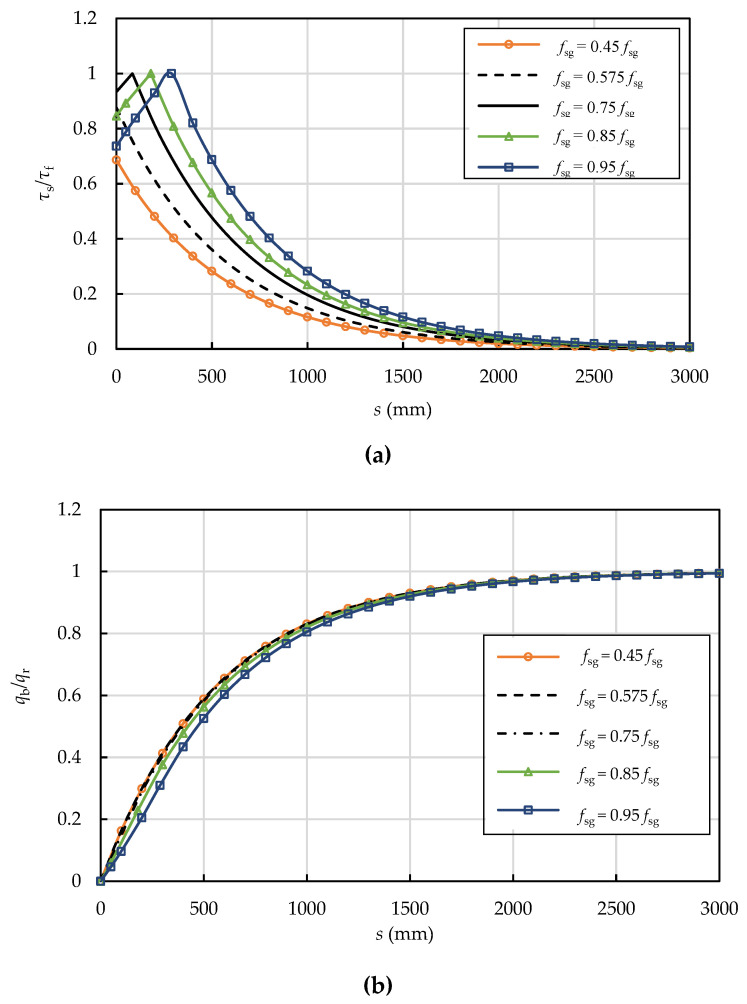
Effect of prestressing level on the (**a**) interfacial shear and (**b**) normal stress distributions (*τ**_f_* = 3.2 MPa, *δ*_1_ = 3.0 mm, *k* = 0.5 and *δ**_f_* = 6.6 mm).

**Figure 13 materials-15-05779-f013:**
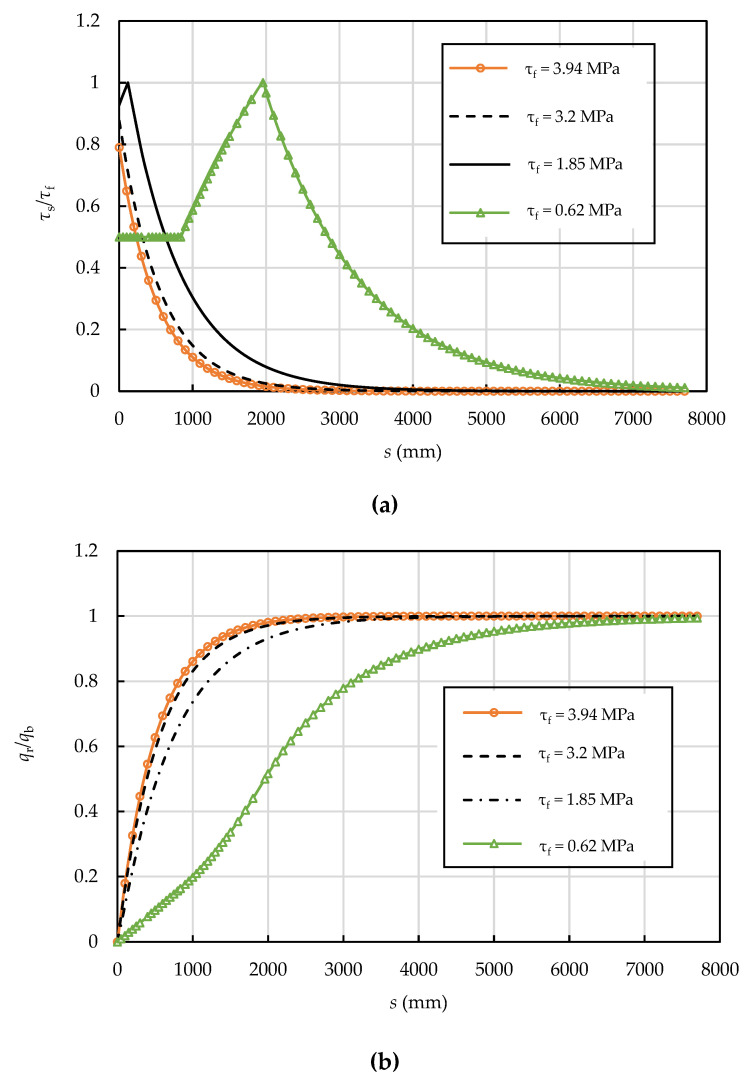
Effect of interfacial shear strength on the (**a**) interfacial shear and (**b**) normal stress distributions (*δ*_1_ = 3.0 mm, *k* = 0.5, and *δ**_f_* = 6.6 mm).

**Figure 14 materials-15-05779-f014:**
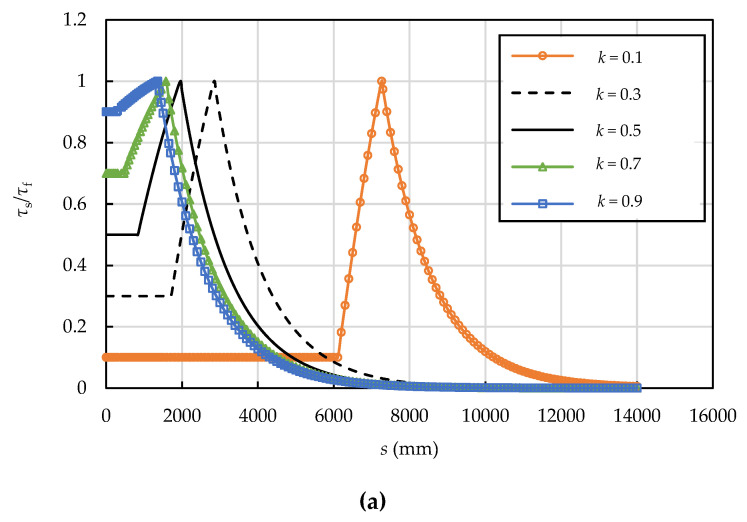
Effect of residual interfacial shear strength factor on the (**a**) interfacial shear and (**b**) normal stress distributions (*τ**_f_* = 0.62 MPa, *δ*_1_ = 3.0 mm, and *δ**_f_* = 6.6 mm).

**Table 1 materials-15-05779-t001:** Material and geometrical parameters of the ECP.

*E**_m_* (GPa)	*E**_s_* (GPa)	*μ* * _m_ *	*f**_cm_* (MPa)	*E**_c_* (GPa)	*μ* * _c_ *	*μ* * _s_ *	*E**_cy_* (GPa)	*μ* * _cy_ *
25.12	193.05	0.17	37.9	42.1	0.18	0.3	206.85	0.3
** *f* ** ** * _sg_ * ** **(MPa)**	** *f* ** ** * _c_ * ** **(MPa)**	** *r* ** ** * _s_ * ** **(mm)**	** *r* ** ** * _m_ * ** **(mm)**	** *R* ** ** * _s_ * ** **(mm)**	** *R* ** ** * _i_ * ** **(mm)**	** *R* ** ** * _cyi_ * ** **(mm)**	** *R* ** ** * _cyo_ * ** **(mm)**	** *R* ** ** * _co_ * ** **(mm)**
902.39	72.5	3.5	25	2350	2000	2089	2091	2350

Note: The specified tensile strength of the prestressing wire *f**_su_* = 1570 MPa.

## Data Availability

The data presented in this study are available on request from the corresponding author.

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
