# Peer review of "An Analytical Solution for Stress Transfer between a Broken Prestressing Wire and Mortar Coating in PCCP"

_materials, 2022, doi:10.3390/ma15165779_

Round 1

Reviewer 1 Report

The article "An analytical solution for stress transfer between a broken pre-2 stressing wire and mortar coating in PCCP" is well written.

·     Abstract need revision with some quantitative results. 

·     Some more latest studies are required in the introduction section to further highlight the importance of this study. 

     The lines 239-240. The mortar coating can provide sufficient anchorage force for the broken wire if the prestressing wire is long enough. Please provide proper reasons for this.

Authors must summarized results in more systematic way with reference to the previous studies.

Conclusions are too limited to proof the significant outcome of this study.

Reviewer 2 Report

The manuscript describes an analytical solution for stress transfer between prestressing strands on a concrete pipe when strands break. The analytical solution is validated with a FE model. The manuscript is well written, scientifically sound, and addresses an important research area. 

The manuscript can be accepted for publication in its present form. 

Reviewer 3 Report

Dear Editor, Dear Authors,

The presented topic is of great practical importance, because prestressed concrete cylinder pipes (PCCPs) are widely used for transporting and distributing water. Based on a tri-linear bond-slip model, the size of prestress loss zone and the stress transfer mechanism between a broken wire and mortar coating are analysed in this paper. Most PCCP failures are related to the breakage of prestressing wires. The load-bearing capacity of PCCP is significantly affected by the size of prestress loss zone and the stress distribution in the broken wire. 

The Authors have presented a shear-lag model for predicting the stress transfer and debonding propagation after a prestressing wire breaks in a PCCP pipe. Based on a parametric study, it has been found that the size of prestressing wire, the prestressing level, the interfacial shear (bond) strength and the residual interfacial shear strength factor have significant effects on the interfacial shear and normal stress distributions. 

However, the authors did not answer whether the presented model is consistent with the actual states of deformations and stresses after the breakage of prestressing wires in a PCCP pipe. 

The model was carefully developed, although it does not take into account the influence of possible imperfections of PCCP pipes. Despite the lack of experimental verification and the omission of the imperfection of pipes, I believe that the article should be published because:

- is the basis for programming an experiment on a full scale;

- concerns an important topic for a large part of society.

Notes for improvement:

1. References [32], [33] is unnecessary. References [34] is an extension of [32] and [33] and does not contain new information.

2. Conclusions should be supplemented with insights relevant to the design and use of PCCP pipes.

Round 2

Reviewer 3 Report

I do not have any more comments.